

# Ginsenoside panaxatriol reverses TNBC paclitaxel resistance by inhibiting the IRAK1/NF-κB and ERK pathways

Panpan Wang[1,2,*], Dan Song[1,*], Danhong Wan[3], Lingyu Li[3], Wenhui Mei[3], Xiaoyun Li[3], Li Han[2], Xiaofeng Zhu[2], Li Yang[1], Yu Cai[1,4] and Ronghua Zhang[1,4]

[1] College of Pharmacy, Jinan University, Guangzhou, China
[2] First Affiliated Hospital of Jinan University, Guangzhou, China
[3] College of Traditional Chinese Medicine, Jinan University, Guangzhou, China
[4] Cancer Research Institute, Jinan University, Guangzhou, China
* These authors contributed equally to this work.

Corresponding authors
Yu Cai, tcaiyu@jnu.edu.cn
Ronghua Zhang, tzrh@jnu.edu.cn

## ABSTRACT

**Background:** Paclitaxel (PTX) resistance is a major obstacle in the treatment of triple-negative breast cancer (TNBC). Previously, we have reported that interleukin-1 receptor-associated kinase 1 (IRAK1) and its downstream pathways are associated with PTX resistance in TNBC cells. In this study, we sought to investigate the combination treatment of ginsenoside panaxatriol (GPT), one of the main active components in *Panax ginseng*, with PTX on viability and apoptosis of TNBC PTX resistant cells, and explore the role of IRAK1 mediated signaling pathways in the therapeutic effects.

**Methods:** CellTiter-Glo and colony formation assays were used to assess cell viability. Flow cytometry was used to analyze subG1 and apoptosis. Western blot was used to detect expressions of proteins involved in apoptosis and the IRAK1/NF-κB and ERK pathways. The mRNA expression of inflammatory cytokines, S100A7/8/9 and cancer stem cell (CSC)-related genes were examined by qPCR. Stem cells were identified by tumor sphere assay. Cell invasion ability was examined by transwell assay.

**Results:** We show that GPT inhibits MDA-MB-231 PTX resistant (MB231-PR) cell viability in a dose-dependent manner. When combined with PTX, GPT synergistically causes more cell death, induces subG1 accumulation and cell apoptosis. Besides, up-regulation of BAX/BCL-2 ratio, and down-regulation of MCL-1 are also observed. Moreover, this combination inhibits IRAK1, NF-κB and ERK1/2 activation, and leads to down-regulation of inflammatory cytokines (IL6, IL8, CXCL1, CCL2), S100A7/9 and CSC-related genes (OCT4, SOX2, NANOG, ALDH1, CD44) expression. In addition, the combination treatment suppresses MB231-PR cell invasion ability, and impairs tumor sphere growth both in MB231-PR and SUM159 PTX resistant (SUM159-PR) cells.

**Conclusion:** Our study demonstrates that GPT can resensitize TNBC PTX resistant cells to PTX by inhibiting the IRAK1/NF-κB and ERK pathways and reducing stem cell characteristics.

## INTRODUCTION

Triple-negative breast cancer (TNBC) is a highly invasive subtype of breast cancer with poor prognosis (*Foulkes, Smith & Reis-Filho, 2010*). Because of the lack of hormone receptors and human epidermal growth factor receptor 2 (HER2) amplification, TNBC does not respond to hormone or anti-HER2 treatment, and mainly relies on traditional chemotherapy (*Denkert et al., 2017*). Paclitaxel (PTX)-based chemotherapy regimens are the most widely used first-line therapeutic strategies for clinically treatment of TNBC. Although effective in the initial treatment, a subset of patients eventually develops resistance, and leads to disease progression (*Mustacchi & De Laurentiis, 2015*; *Schettini et al., 2016*). Hence, it is highly necessary to find a solution for PTX resistance in TNBC.

The nuclear factor kappa B (NF-κB) signaling pathway plays an important role in cancer initiation, progression and resistance, thus making it a good target for cancer treatment (*Chaturvedi et al., 2011*; *Hoesel & Schmid, 2013*; *Taniguchi & Karin, 2018*). However, despite numerous attempts to develop molecular drugs that specifically target NF-κB, few clinical advancements have been made (*Baud & Karin, 2009*). Previously, by using gain and loss of function methods, we reported that activation of interleukin-1 receptor-associated kinase 1 (IRAK1), an upstream kinase of the NF-κB signaling pathway, is associated with PTX resistance in TNBC cells (*Wee et al., 2015*). Importantly, together with S100A7, S100A8 and S100A9 (S100A7/8/9), IRAK1 form a druggable circuitry which drives the malignancy of TNBC cells (*Goh et al., 2017*). These observations prompted us to search for potential candidate drugs that can target IRAK1 and its downstream signaling pathways.

Ginseng and its active ingredient ginsenosides, such as ginsenosides Rg3 (GRg3), have been widely used in China to treat cancers in the clinic. Ginsenosides are a class of steroid glycosides and triterpene saponins. Over the last decade, more than 100 different types have been isolated and identified. Researchers have found that GRg3 can facilitate the penetration of PTX through the Caco-2 monolayer from the apical side to the basal side, and enhance the oral bioavailability of PTX in vivo (*Yang et al., 2012*). Furthermore, GRg3 can inhibit P-glycoprotein expression and increase the accumulation of drugs such as vincristine in multidrug resistant cells, but not in sensitive cells (*Kim et al., 2003*). Importantly, it has been reported that some ginsenosides can inhibit the activation of IRAK1 and its downstream pathways (*Joh et al., 2011*; *Nag et al., 2012*; *Shaukat et al., 2019*). In this study, we investigated the in vitro anti-viability of ginsenoside panaxatriol (GPT) in TNBC PTX resistant cells, and found that GPT can target IRAK1/NF-κB and ERK pathways to overcome resistance.

## MATERIALS AND METHODS

### Chemicals and reagents

Ginsenoside panaxatriol was obtained from Must Bio-Technology (Chengdu, China). PTX was purchased from Sigma-Aldrich (St. Louis, MO, USA). Dulbecco's Modified Eagle

Medium (DMEM) (11995-040), F-12 nutrient mixture (Ham) and fetal bovine serum (FBS) were bought from Life Technologies (Grand Island, NY, USA). MammoCul medium (human) and supplements were purchased from STEMCELL Technologies (Vancouver, BC, Canada). CellTiter-Glo luminescent cell viability assay kits were purchased from Promega Corporation (Madison, WI, USA). iScript gDNA Clear cDNA Synthesis Kits and iTaq Universal SYBR Green Supermix Kits were purchased from Bio-Rad Laboratories (Hercules, CA, USA). p-IRAK1 S376, IRAK1, p-P65 S536, P65, p-ERK1/2, ERK1/2, BAX, BCL-2 and MCL-1 antibodies were supplied by Cell Signaling Technology (Danvers, MA, USA). Beta-actin antibody was purchased from Sigma–Aldrich (St. Louis, MO, USA).

## Cell culture and viability assay

MDA-MB-231 cells and SUM159 cells were obtained from ATCC. MDA-MB-231 PTX resistant (MB231-PR) cells and SUM159 PTX resistant (SUM159-PR) were established as previously described (*Wee et al., 2015*). Briefly, cells were treated with increasing concentrations of PTX for over a period of 3 months. Then, MB231-PR cells were cultured in DMEM supplemented with 75 nM PTX, 1% penicillin/streptomycin, and 10% FBS at 37 °C with 5% $CO_2$. SUM159-PR cells were maintained in F-12 supplemented with 300 nM PTX, 5% FBS, 10 mM HEPES, 10 μg/ml hydrocortisone, 5 μg/ml insulin and 1% penicillin/streptomycin. For cell viability assay, 1,000 cells/well in 90 μl medium were seeded into Costar 96-well white plates. The next day, different concentrations of drugs in 10 μl medium were added and incubated for the indicated times. Then, cells were lysed with 50 μl CellTiter-Glo reagent and the chemiluminescent signals were detected with a PerkinElmer VICTOR X4 plate reader.

## Cell cycle and apoptosis assay

Cell cycle and apoptosis analysis was performed by DNA content quantification to quantify the subG1 population, which is a reflective of the extent of cell death. Briefly, floating and adherent cells were harvested together after 48 h and 72 h treatment respectively. Then, cells were fixed by 70% ethanol at 4 °C overnight. After washing with phosphate buffered saline (PBS), cells were resuspended in 100 μl of 100 μg/ml RNase A. 5 min later, 400 μl of 50 μg/ml propidium iodide was added, and cells were incubated for 30 min in dark area. Finally, the stained cells were analyzed by FACScalibur and quantified using CellQuest software.

## Colony formation assay

A total of 1,000 cells/well were seeded into 12-well plates. The next day, drugs were added and incubated for 12 days. Medium was changed every 3 days. Then, cells were washed with PBS and fixed with methanol for 10 min. Finally, cells were stained with 0.1% crystal violet at room temperature for 10 min and photographed.

## Tumor sphere formation assay

A total of 3,000 cells/well in 180 μl medium were seeded into Corning 96-well spheroid microplates in complete MammoCul medium. The next day, drugs in 20 μl medium were added and incubated for 12 days. Pictures were taken on day 6 and day 12. Finally, on

**Table 1 qPCR primers.**

| Gene | Forward (5′–3′) | Reverse (5′–3′) |
| --- | --- | --- |
| IL6 | AGTTCCTGCAGAAAAAGGCAAAG | AAAGCTGCGCAGAATGAGAT |
| IL8 | ACCGGAAGGAACCATCTCAC | GGCAAAACTGCACCTTCACAC |
| CXCL1 | CCAGCTCTTCCGCTCCTC | CACGGACGCTCCTGCTG |
| CCL2 | CCCAAAGAAGCTGTGATCTTCA | TCTGGGGAAAGCTAGGGGAA |
| S100A7 | GACAAGATTGAGAAGCCAAGCC | TGTGCCCTTTTTGTCACAGG |
| S100A8 | TGCCGTCTACAGGGATGAC | TCTGCACCCTTTTTCCTGATATAC |
| S100A9 | TCCTCGGCTTTGACAGAGTG | TGGTCTCTATGTTGCGTTCCA |
| OCT4 | CTGGGTTGATCCTCGGACCT | CCATCGGAGTTGCTCTCCA |
| SOX2 | GCCGAGTGGAAACTTTTGTCG | GGCAGCGTGTACTTATCCTTCT |
| NANOG | TTTGTGGGCCTGAAGAAAACT | AGGGCTGTCCTGAATAAGCAG |
| ALDH1 | CTGCTGGCGACAATGGAGT | GTCAGCCCAACCTGCACAG |
| CD44 | TGCCGCTTTGCAGGTGTATT | CCGATGCTCAGAGCTTTCTCC |
| 18S | CGAACGTCTGCCCTATCAACTT | ACCCGTGGTCACCATGGTA |

day 12, cells were lysed with 100 µl CellTiter-Glo reagent and the chemiluminescent signal was detected with a PerkinElmer VICTOR X4 plate reader.

## Transwell invasion assay

A total of 10,000 cells in 100 µl serum-free DMEM containing DMSO, PTX, GPT, or combination were added into Corning Transwell polycarbonate membrane inserts coated with Matrigel (300 µg/mL). And medium containing 10% FBS was added to the bottom chamber. After 24 h incubation, the cells that remained on the above surface of the insert membrane were scraped off with a cotton swab. The cells that passed through Matrigel to the bottom of the insert were fixed with paraformaldehyde and stained with 0.1% crystal violet in methanol. The inserts were photographed, and the cells were counted.

## Quantitative-PCR (qPCR) assay

RNA extraction and purification were performed according to the instructions from Zymo Research (R2052). A total of 750 ng RNA was used to synthesize cDNA. And qPCR was performed using the Applied Biosystems 7500 Fast Real-Time PCR system. All primers are listed in Table 1. For quantification of mRNA levels, 18S was used as the internal control, and the expression of target genes were analyzed using the $2^{-\Delta\Delta Ct}$ method.

## Western blot assay

Western blot was performed using whole-cell extracts in protein lysis buffer with freshly added protease inhibitor cocktail. Proteins were separated on 8–10% SDS polyacrylamide gel electrophoresis gels and transferred to polyvinylidene difluoride membranes. The membrane was blocked with 5% non-fat dry milk in tris-buffered saline (TBS) containing 0.1% Tween 20 (TBST). The membrane was then incubated with primary

antibody (1:1,000 dilution) in 5% bovine serum albumin overnight. After washed three times with TBST, the membrane was incubated with secondary antibody (1:2,000 dilution) in 5% non-fat dry milk at room temperature for 1 h. Then, SuperSignal West Femto Maximum Sensitivity Substrate was added, and images were taken using the ChemiDoc MP System.

## Statistical analysis

Data are shown as mean ± SD. The t test was used to determine whether there are any statistically significant differences between two groups. $P < 0.05$ was considered statistically significant.

# RESULTS

## GPT promotes cytotoxicity of PTX in MB231-PR cells

To explore whether GPT can promote cytotoxicity of PTX in TNBC resistant cells, MB231-PR was constructed and used as cell model. Firstly, we conducted CellTiter-Glo assay to observe different concentration of GPT on cell viability. As shown in Fig. 1A, GPT treatment significantly decreased cell viability of MB231-PR cells in a dose dependent manner, with the half maximal inhibitory concentration (IC50) 21.39 µM. Secondly, we combined GPT with PTX to check whether they have synergistic effects. Results showed that the combination caused dramatic cell death in a dose and time dependent manner, comparing to either single use group (Fig. 1B). Interestingly, the synergistic effects didn't apply to MB231 parental (MB231-PT) cells, although MB231-PT cells were sensitive to PTX (Fig. S1) and showed more sensitive to GPT when treated with the same concentration (Fig. 1C). Notable, the clinical using drug GRg3 didn't cause significant cell death in single or combination treatment group (Fig. S2). In addition, colony formation assay confirmed the synergistic cytotoxicity effects of the combination on MB231-PR cells (Fig. 1D; Fig. S3).

Since chemotherapy resistance appears partly due to aberrant changes of signaling pathways that endowed cells with the abilities to escape apoptosis, restoring apoptosis is a very important therapeutic strategy for antitumor therapy (*Baig et al., 2016*; *Plati, Bucur & Khosravi-Far, 2008*). Therefore, next, we used flow cytometry to measure subG1 changes after the combination treatment, which is marker of apoptosis. Not surprisingly, GPT combined with PTX significantly increased subG1 cell accumulation both after 48 h and 72 h (Fig. 1E; Fig. S4). Taken together, these results suggested GPT as a very effective molecular to reverse PTX resistance in TNBC cells.

## The combination treatment activates mitochondria mediated apoptosis

The alteration of pro-apoptotic proteins and anti-apoptotic proteins play important roles in the determination of cancer cells apoptosis, and are associated with chemoresistance (*Campbell & Tait, 2018*; *Warren, Wong-Brown & Bowden, 2019*). Thus, we observed the protein expression of BAX and BCL-2 after treatment, two key mediators of apoptotic response to chemotherapy. As shown in Figs. 2A and 2B, GPT combined with PTX

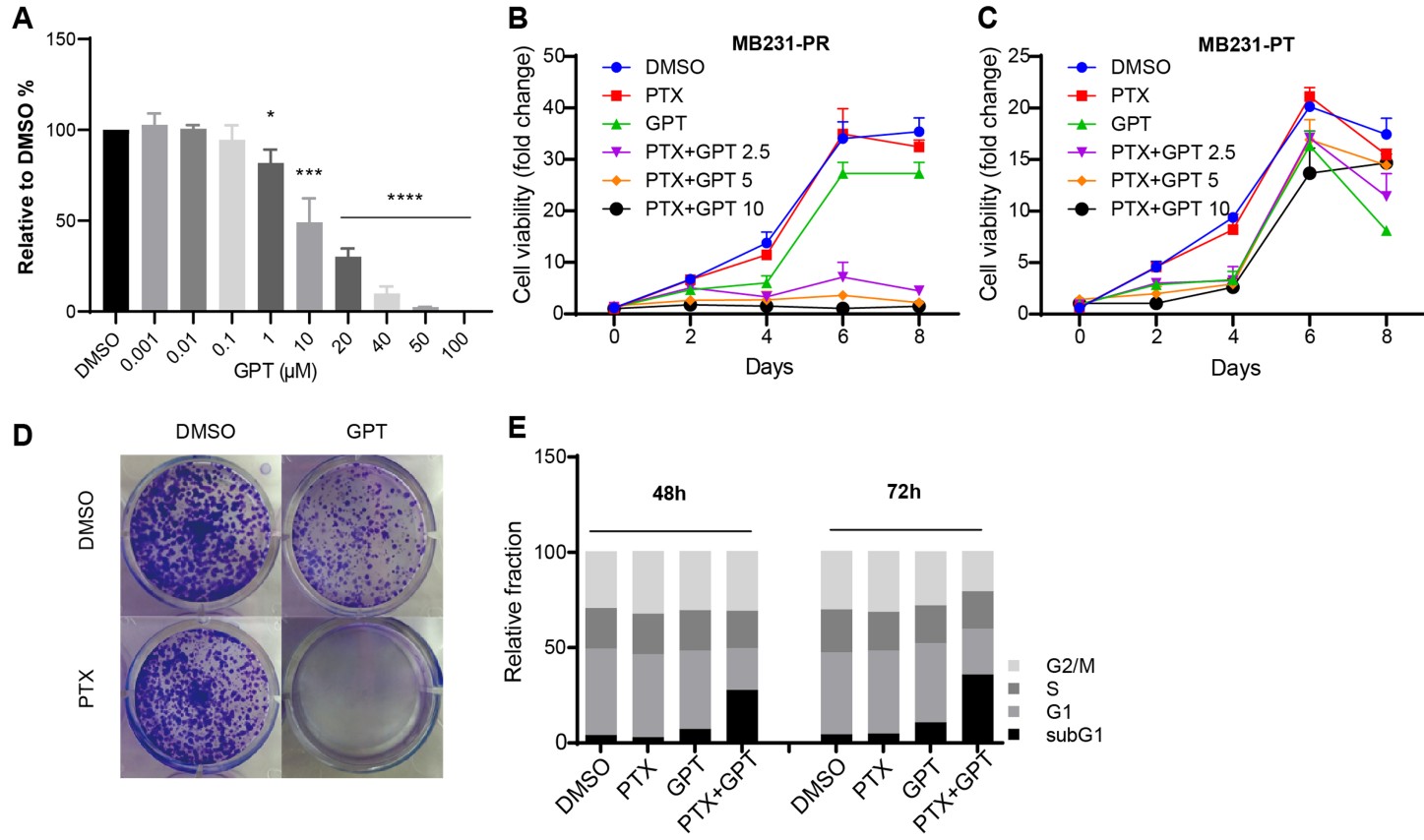

**Figure 1 GPT combined with PTX inhibit MB231-PR cell viability and induce cell apoptosis.** (A) Single treatment of GPT on MB231-PR cell viability. Cells were treated with different concentration of GPT for 4 days. (B) Combination treatment of GPT and PTX on MB231-PR cell viability. Cells were treated with DMOS, 75 nM PTX, 10 μM GPT, 75 nM PTX + 2.5 μM GPT, 75 nM PTX + 5 μM GPT, 75 nM PTX + 10 μM GPT, respectively. (C) Combination treatment of GPT and PTX on MB231-PT cell viability. Cells were treated with DMSO, 1 nM PTX, 10 μM GPT, and different combination, respectively. (D) Representative images of colony formation assay. MB321-PR cells were treated for 12 days with DMSO, 75 nM PTX, 10 μM GPT and combination, respectively. (E) Flow cytometry detection of cell cycle after treatment for 48 h and 72 h. $^*P < 0.05$, $^{***}P < 0.001$, $^{****}P < 0.0001$. P-values were calculated with $t$ test.

significantly increased BAX and decreased BCL-2 expression in a dose and time dependent manner.

Besides BAX and BCL-2, MCL-1 was recently reported to be associated with poor prognosis in TNBC patients and can be used as a therapeutic target (*Campbell et al., 2018*). Notably, we have shown that IRAK1 inhibitor can decrease MCL-1 expression in MB321-PR cells to induce cell apoptosis (*Wee et al., 2015*). Therefore, we also evaluated the protein expression of MCL-1 after treatment. As shown in Figs. 2A and 2B, the combination treatment also resulted in down-regulation of MCL-1 expression. These results together suggested that the combination treatment activated mitochondria mediated apoptosis to reverse PTX resistance.

### The combination treatment inhibits IRAK1/NF-κB and ERK pathways

To further clarify the signaling pathways that involved in GPT effects, gene expression profiling was conducted in MB231-PR cells treated with DMSO, PTX, GPT and combination, respectively. Results showed that NOD-like receptor signaling pathways

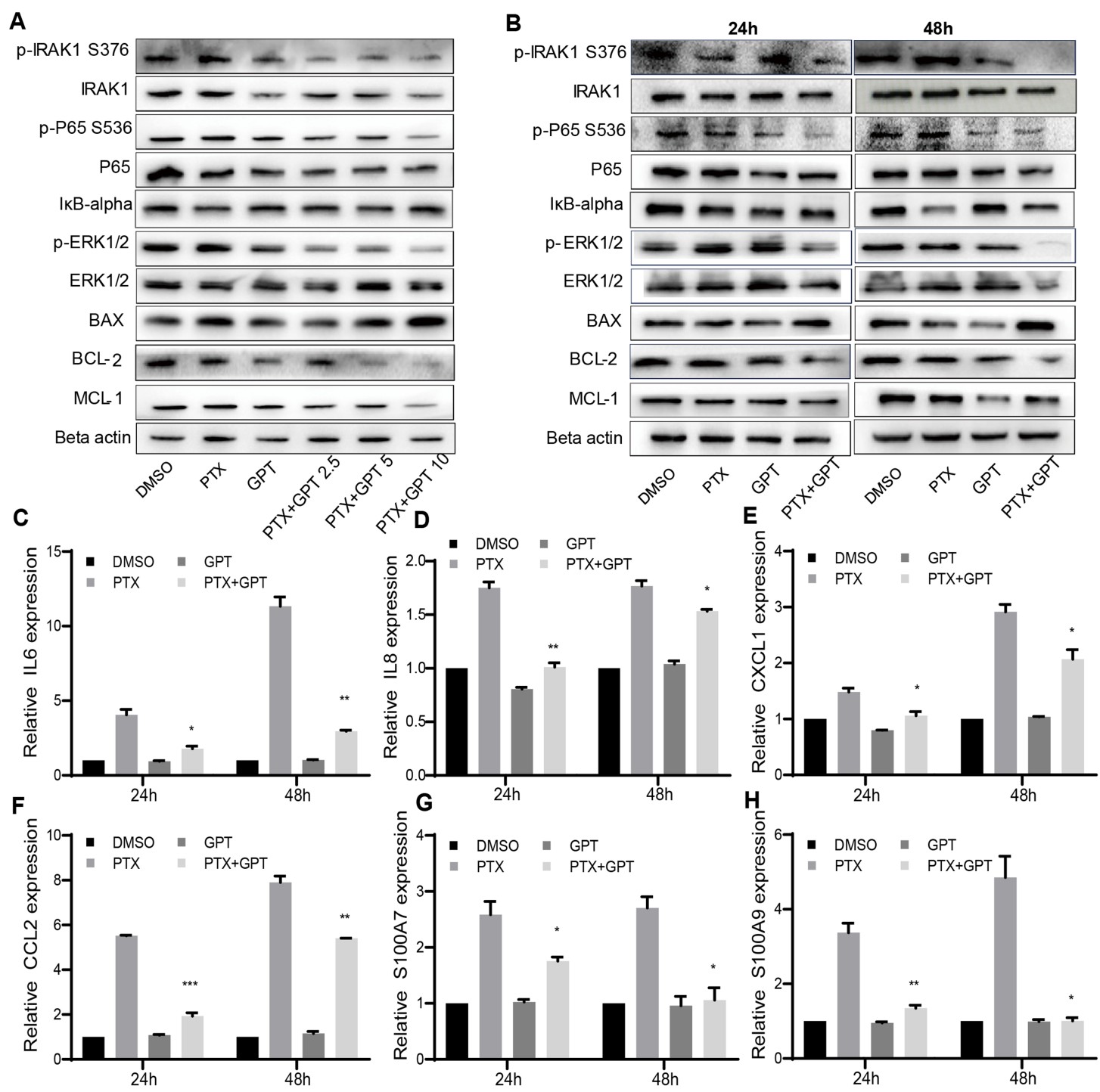

**Figure 2 The combination treatment activates apoptosis pathway and inhibits IRAK1/NF-κB, ERK pathways in MB231-PR cells.** (A) Western blot analysis of proteins expression after cells treated with DMSO, 75 nM PTX, 10 μM GPT and different combination for 24 h. (B) Western blot analysis of proteins expression after cells treated with DMSO, 75 nM PTX, 10 μM GPT and combination for 24 h and 48 h, respectively. (C–H) qPCR analysis of IRAK1/NF-κB downstream inflammatory cytokines and S100A7/9 gene expression after cells treated for 24 h and 48 h, respectively. $*P < 0.05$, $**P < 0.01$, $***P < 0.001$. $P$-values were calculated with $t$ test.

played an important part in GPT activity in MB231-PR cells (data not shown). Interestingly, through loss and gain of function study, we have previously reported that activation of IRAK1, a key kinase of NOD-like receptor signaling pathway, is associated with PTX resistance in TNBC cells (*Wee et al., 2015*) Moreover, target IRAK1 using pharmacologic inhibitor can induce MB231-PR cells apoptosis, when combined with PTX (*Wee et al., 2015*). Thus, consideration was given to IRAK1 and its downstream signaling pathways. Results showed that the combination treatment can significantly inhibit the phosphorylation of IRAK1, P65, ERK1/2, and increase the expression of IκB-alpha in a dose and time dependent manner (Figs. 2A and 2B).

To additionally characterize the functional effects of IRAK1 mediated pathways, we investigated the mRNA expression of NF-κB target genes by qPCR, including interleukin 6 (IL6), IL8, chemokine (C-X-C motif) ligand 1 (CXCL1), and chemokine (C-C motif) ligand 2 (CCL2). The above cytokines were shown to be distinctly expressed among different group in our gene expression profiling experiment, and were reported to be critical for the anchorage independent growth of TNBC cells (*Hartman et al., 2013*). As shown in Figs. 2C–2F, (Tables S1 and S3), compared to DMSO, PTX significantly promoted the expression of IL6, IL8, CXCL1 and CCL2. However, this induction can be significantly attenuated when combined with GPT.

Except these target cytokines, we previously published that IRAK1 and S100A7/8/9 formed a feedback loop to drive the malignancy of TNBC cells (*Goh et al., 2017*). Here, we also showed that the combination treatment significantly decreased S100A7 and S100A9 mRNA expression (Figs. 2G and 2H; Tables S1 and S3), although S100A8 mRNA expression level was too low to be detected. These results together suggested that the combination treatment overcome PTX resistance by inhibiting IRAK1 mediated NF-κB and ERK pathways.

## The combination treatment inhibits cancer stem cell-related genes expression and impairs tumor sphere growth and invasion ability

Accompanied with killing cancer cell, PTX treatment has been reported to induce cancer stem cell (CSC) enrichment, another key mechanism suggested to be responsible for chemoresistance and cancer metastasis (*Bousquet et al., 2017*; *Zhang et al., 2019*). And drug that can target cancer stemness are proposed as new strategies for clinical cancer treatment (*Saygin et al., 2019*; *Sun et al., 2019*). In order to testify the effect of combination therapy on characteristics of CSC, firstly, qPCR was used to check the expression of a group of CSC-related genes. As shown in Figs. 3A– 3E, (Tables S2, S4 and S5), compared to PTX, the combination treatment significantly lead to down-regulation of octamer-binding transcription factor 4 (OCT4), sex determining region Y-box 2 (SOX2), NANOG, aldehyde dehydrogenase 1 (ALDH1), and CD44 gene expression. Secondly, transwell invasion and tumor sphere assay were conducted to assess CSC properties. As shown in Figs. 3F and 3G–3J, the combination treatment significantly suppressed MB231-PR cell invasion ability, and impaired tumor sphere growth both in MB231-PR and SUM159-PR cells.

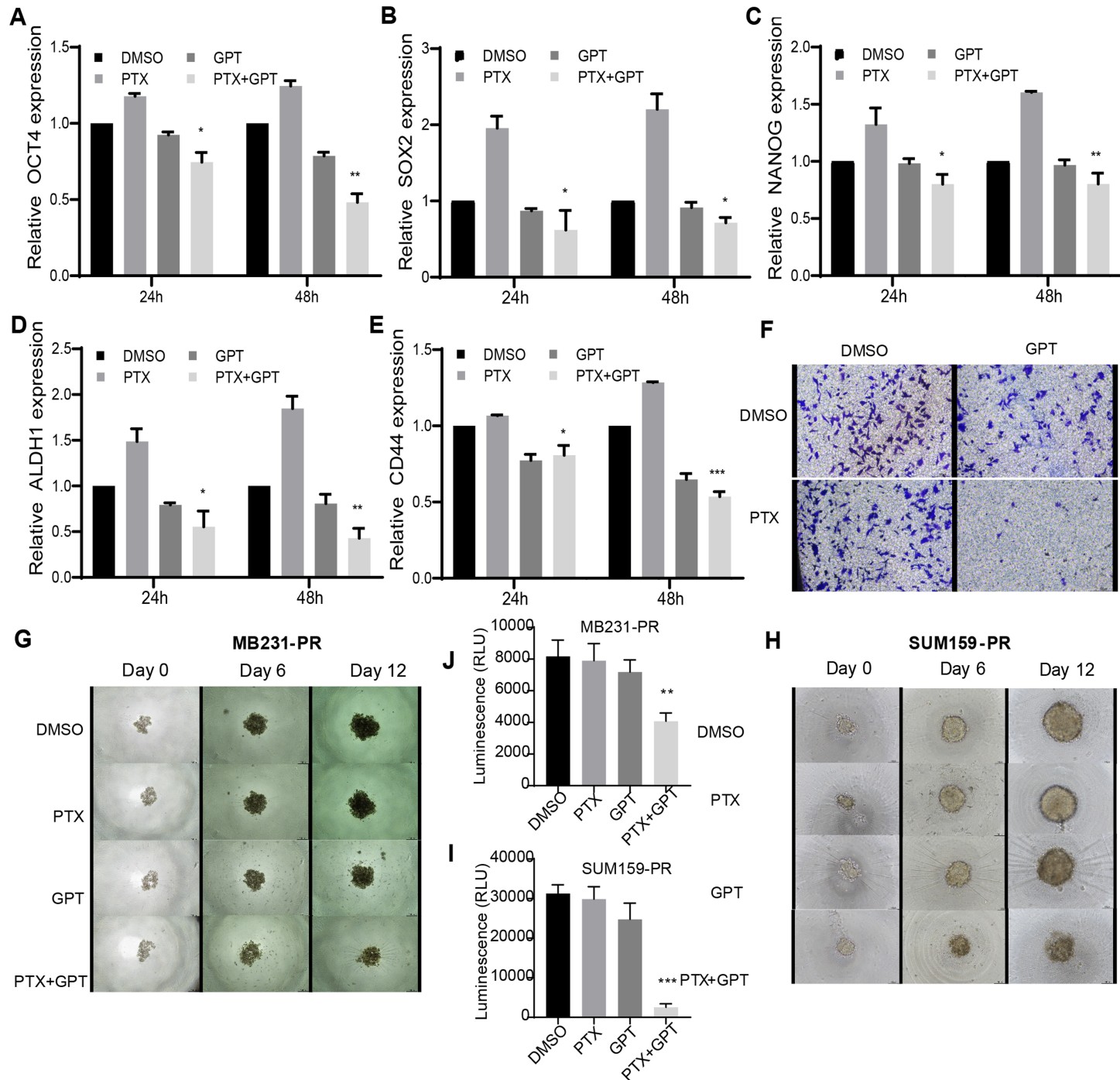

**Figure 3 The combination treatment inhibits inflammatory cytokines expression, tumor sphere growth and cell invasion ability.** (A–E) qPCR analysis of CSC-related genes expression after cells treated with DMSO, 75 nM PTX, 10 μM GPT, and combination for 24 h and 48 h, respectively. (F) Transwell invasion assay of MB231-PR cells after drug treatment. Cells were seeded into Corning transwell polycarbonate membrane inserts coated with Matrigel (300 μg/mL) and cultured for 24 h. (G–J) Representative images of tumor sphere assays. Cells were seeded into Corning 96-well spheroid microplates and cultured with MammoCul medium. Tumor sphere was observed after treated for 12 days. *P < 0.05, **P < 0.01, ***P < 0.001. P-values were calculated with t test.

## DISCUSSION

Treatment of TNBC has been challenging, due to lack of target therapy options and constantly acquired resistance. Therefore, a combinatorial therapy has been always preferred to achieve a synergistic effects. Plant derived compounds, such as saponins, flavonoids and alkaloids, have been tested and proved to be effective in killing cancer cells and restoring resistant cells to chemotherapy (*Aung et al., 2017*). Of which, ginsenosides have been researched in different cancers. Results showed that GRg3 can enhance the anti-cancer ability of chemo drugs by modulating the oral bioavailability (*Yang et al., 2012*), inhibiting P-glycoprotein expression (*Kim et al., 2003*), inhibiting cell autophagy (*Wang et al., 2019*), and down regulating epidermal growth factor receptor (EGFR)/ phosphatidylinositol-3-kinase (PI3K)/ Akt signaling pathway (*Jiang et al., 2017*). In this study, we investigated the combination treatment of GPT and PTX on viability and apoptosis of TNBC PTX resistant cells, and clarified the signaling pathways underlies. Our data showed that the combination can synergistically inhibit MB231-PR cell viability, induce subG1 accumulation and trigger the mitochondrial mediated apoptosis. Our data further suggested that the combination can inhibit IRAK1/NF-κB and ERK signaling pathways, resulted in down-regulation of inflammatory factors and S100A7/9 expression, which are the main cytokines in tumor microenvironment contributed to CSC phenotype and function. In addition, we showed that combination can inhibit CSC-related genes expression and impair invasion ability and tumor sphere growth.

It is suggested that the BCL-2 family are key mediators of anti-cancer therapeutics, and abnormal expression of apoptotic proteins contributed to chemoresistance (*Hata, Engelman & Faber, 2015*). In addition to other members, decreased BAX/BCL-2 ratio and elevated MCL-1 expression were reported to be closely related with PTX resistance in breast cancer (*Lee et al., 2017*; *Sharifi et al., 2014*). Drugs which can inhibit the activity of these proteins are believed to improve the efficacy of chemotherapeutic agents. Interestingly, our data showed that GPT augments the effects of PTX by up-regulating BAX/BCL-2 ratio and down-regulating MCL-1 expression.

The results in this study are consistent with our previous published papers, showing that pharmacologic inhibition of IRAK1 phosphorylation and downstream signaling pathways activation can overcome TNBC PTX resistance. Notably, other group recently reported that the expression of IRAK1 was positively correlated with tumor size following neoadjuvant chemotherapy (NCT) (*Yang et al., 2019*). Breast cancer patients, with higher expression of IRAK1 both before and after NCT, had a shorter survival period (*Yang et al., 2019*). These results together highlight the role of IRAK1 in chemoresistance and clinical application of IRAK1 inhibitors. IκB-alpha is a downstream kinase of IRAK1. It has been reported that IκB-alpha plays an important role in NF-κB cytosolic-nuclear translocation. IκB-alpha enters the nucleus to bind NF-κB dimers and translocate them to the cytosol (*Christian, Smith & Carmody, 2016*). Researchers also showed that IκB-alpha was the key mediator responsible for PTX induced NF-κB nuclear translocation, DNA binding and transcriptional activity (*Huang et al., 2000*). Consistently, decreased IκB-alpha and increased NF-κB transcriptional activity after PTX

treatment can also be seen in our experiment. However, the combination treatment increased IκB-alpha expression and decreased NF-κB transcriptional activity.

Another our major finding is that inhibition of IRAK1/NF-κB and ERK pathways by GPT reduced stem cell characteristics. CSCs have been reported as one of the determining reasons for chemoresistance and subsequent cancer relapse. And one of the mechanisms that CSCs are acquired is taking advantage of PTX treatment induced inflammation cytokines and S100 protein family in tumor microenvironment.

In our experiment, decreased expression of inflammation cytokines (IL-6, IL-8, CXCL1 and CCL2) can be noticed in the combination group. The above cytokines are reported to be NF-κB transcriptional targets, and their expression are induced following NF-κB activation after chemo treatment (*Jia et al., 2017*). In turn, these factors activate inflammation related signaling pathways such as NF-κB and signal transducer and activator of transcription 3 (STAT3) (*Wang et al., 2018*; *Wong, Che & Leung, 2015*; *Yue et al., 2006*), which further promote cell survival through regulating apoptosis proteins and promote the formation of CSC through regulating CSC related genes (*Rhyasen et al., 2013*). Importantly, in accordance to IL-8 inhibitor, anti-IL6 antibody, anti-CXCL1 antibody, or anti-CCL2 antibody, here we showed that target IRAK1 mediated pathways by GPT can effectively down-regulate these cytokines and disrupt this process (*Dey, Rathod & De, 2019*; *Heo, Wahler & Suh, 2016*; *Miyake et al., 2019*; *Teng et al., 2017*).

Besides, we also identified that S100A7/9 were down-regulated after combination treatment. S100A7/9 are members of the S100 protein family, which are closely related to tumorigenesis and progression (*Cancemi et al., 2018*; *Chen et al., 2014*). In addition, S100A7/8/9 can be regulated by NF-κB and STAT3, which in turn can activate NF-κB and ERK (*Hermani et al., 2006*; *Liu et al., 2013*; *Nemeth et al., 2009*). S100A8/9 and CXCL1/2, or S100A7/8/9 and IRAK1, form a feedback loop to cause cancer chemoresistance and drive breast cancer tumor sphere growth (*Acharyya et al., 2012*; *Goh et al., 2017*). Collectively, our data suggested that GPT can disrupt this feedback loop to inhibit CSC characteristics.

As to molecular phenotype in breast cancer, CSCs display CD44+/CD24- phenotype and high ALDH1 activity. In parallel, other characters include overexpression of transcription factors OCT4, SOX2 and NANOG, which are associated with high-grade stage and poor clinical outcome in TNBC. In this part, we demonstrated that GPT combined with PTX can inhibit CSCs related gene expression, impair invasion ability and tumor sphere growth.

## CONCLUSIONS

Our study demonstrates that GPT can resensitize TNBC PTX resistant cells to PTX treatment by inhibiting the IRAK1/NF-κB and ERK pathways, reducing stem cell characteristics, thus provide it as a novel molecular for clinic use.

## ACKNOWLEDGEMENTS

We thank all the reviewers for their helpful comments.

### Funding

This work was supported by the National Natural Science Foundation of China (No. 81603342) and the Administration of Traditional Chinese Medicine of Guangdong Province (No. 20171074). The funders had no role in study design, data collection and analysis, decision to publish, or preparation of the manuscript.

### Grant Disclosures

The following grant information was disclosed by the authors:
National Natural Science Foundation of China: 81603342.
Administration of Traditional Chinese Medicine of Guangdong Province: 20171074.

### Competing Interests

The authors declare that they have no competing interests.

### Author Contributions

- Panpan Wang conceived and designed the experiments, performed the experiments, analyzed the data, prepared figures and/or tables, and approved the final draft.
- Dan Song performed the experiments, analyzed the data, prepared figures and/or tables, and approved the final draft.
- Danhong Wan analyzed the data, authored or reviewed drafts of the paper, and approved the final draft.
- Lingyu Li analyzed the data, authored or reviewed drafts of the paper, and approved the final draft.
- Wenhui Mei analyzed the data, authored or reviewed drafts of the paper, and approved the final draft.
- Xiaoyun Li analyzed the data, authored or reviewed drafts of the paper, and approved the final draft.
- Li Han analyzed the data, authored or reviewed drafts of the paper, and approved the final draft.
- Xiaofeng Zhu analyzed the data, authored or reviewed drafts of the paper, and approved the final draft.
- Li Yang analyzed the data, authored or reviewed drafts of the paper, and approved the final draft.
- Yu Cai conceived and designed the experiments, authored or reviewed drafts of the paper, and approved the final draft.
- Ronghua Zhang conceived and designed the experiments, authored or reviewed drafts of the paper, and approved the final draft.

### Data Availability

The raw measurements are available in the Supplemental Files.

## Supplemental Information

Supplemental information for this article can be found online at http://dx.doi.org/10.7717/peerj.9281#supplemental-information.

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
