# Peer review of "Ginsenoside panaxatriol reverses TNBC paclitaxel resistance by inhibiting the IRAK1/NF-κB and ERK pathways"

_PeerJ, doi:10.7717/peerj.9281_

## Round 0.1 · original submission · Major Revisions

Please carefully address all the issues raised by the referees.

Reviewer 1 ·

Basic reporting

Authors study the role of a ginseng related molecule in sensitisazion of paclitaxel resistant cells to drug treatment.
The topic is important, the english is acceptable in the text but legends are poorly understandable, background can be made clearer, in particular stressing why the ginseng derived molecule is making breast cancer cells more sensitive.
The results are relevant to the hypothesis, however.
Some items to be addressed:
1) Ginsenoside PT has to be briefly described also in the abstract
2) Ginsenoside family and structure has to be summarized in the introduction. For instance I found in another paper this description: "The ginsenosides are the major active pharmacological components of ginseng. Ginsenosides, also known as steroid-like saponins, are unique to ginseng species. There are more than 100 ginsenosides etc.
Abundant literature is available on several ginsenosides and it sould be expanded.
3) The production of the resistant cell line has to be described, not only cited. Only one line is employed.
Raw data are shared but the definition of "Full-lenght uncropped blots" is erroneous, since what presented are portions of western blots and not the entire one. Please provide.

Experimental design

The research is experimental and within the scope of the journal.
Figures are not easilyy readable. Since Paclitaxel is abbreviated as (PTX) I suggest that Ginsenoside PT is abbreviated with a different acronym than PT, because it makes the figures very confusing (GPT? GiPT?). The abbreviations are never used in the text, I suggest that after the first mentioning of ginsenoside PT the abbreviation is used for faster reading.
In Fib 1 C at day 8 a rather catastrophic fall of cell viability occurs with GPT. The explanation given in the results, is not satisfactory. How is that the parental cells are even more sensitive to GPT than to PTX? those compounds are supposed to have a limited toxicity. I suggest to repeat the experiment or to come up with a molecular explanation.
Methods, besides the preparation of the resistant cell lines, are sufficient.
There is no control sensitizing drug, this is a major limit, so a second molecule, already known for sensitization of PTX resistant clones should be tested at least in some experiments

Validity of the findings

The findings are valid, but they would appear more novel when the Ginsenoside PT is compared to other well studied ones.
Data are sound, with the exception of the viability assay.
Discussion should include a part on the use of plant derived preventive drugs for sensitisation to chemotherapy as general concepts as well as other pathways addressed by ginsenosides should be mentioned (for instance PMID: 32067911.
PubMed PMID: 30772396. PubMed PMID: 32020217. PMID: 29035827 etc)

Additional comments

The paper is approaching an interesting topic. In the future I think in vivo studies would be necessary.
More resistant cell lines should be also employed
At least a comparison with a different agent is required.

·

Basic reporting

No comment.

Experimental design

No comment.

Validity of the findings

No comment.

Additional comments

The authors analyzed to the effects of ginsenoside PT as a molecule to revert paclitaxel resistance in an aggressive breast cancer cell line model. The manuscript is potentially interesting and the results valuable for readers in the field. However, some issues need to be adjusted before final acceptance.
My major concern is based on the results in Figure 1, particularly in Panel 2B and 2C. MB231-PR cells have been created to be resistant to paclitaxel. But from Figure 1C, it is evident that MB231-PT cells (parental) seem not to be sensitive to paclitaxel at all. How can we accept the validity of these observations and the following ones if parental cells are not sensitive to paclitaxel??? This needs to be clearly addressed by the authors.
The second major comment refers to Figure 2: to demonstrate the effect of combination treatment on IRAK1 and NF-kB pathways, qPCR has been performed. However, the effects rely on the fact that PTX increases these two signaling pathways (as seen in qPCR experiments), even if at protein level p-IRAK1 and p-p65 did not increase in response to PTX (both panels 2A-2B). The authors need to convince the reviewers these two main mediators are differentially regulated in response to PTX; for instance, given p65 and generally NF-kB proteins shuttle between cytosol and nucleus, maybe the level of phosphorylation does not change, but the localization does it to stimulate the expression of downstream targets. Also, other pathways can converge on these regulations (i.e., STAT3). Please clarify this.
Third: the authors have used only MDA-MB-231 as TNBC cellular model. To better generalize their results, at least one set of the experiments should also be confirmed in another model for TNBC, such as for example MDA-MB-468, BT-459, or Hs578T cells among others.

Minor issues:

- There are some typos throughout the manuscript that need to be fixed (i.e., "resistant" instead of "resistance" line 148; missing full stops; few words misused).

- Methods and Results sections are correctly subdivided in paragraphs, but with a useless numbering. This is not needed, and it would have been more coherent to use it for the whole manuscript, not only in 2 out of the different sections. Please remove it.

- The authors abbreviated the "MDA-MB-231" cell line to "MB1231" throughout the manuscript for simplicity. Given this is not the official name of the cell line, at least once (preferentially in the methods section) they need to use the official name.

- According to the methods, it seems the authors did not use Glutamine in their cultures; however, it is an essential supplement I'm pretty sure has been added to the culture medium. Please revise this.

- There are some problems with supplementary data... There are 2 Word documents... S1 and S2... But each of them names Fig. S1 and Fig. S2. This is confusing.
Moreover, in the first paragraph of Results (lines 152-153) the authors refer to apoptosis induction due to the combination treatment indicating Fig. 1E and Fig. S1 panels. However, while Fig. 1E shows sub G1 and other cell cycle fractions, in Fig. S1 there is not any FACS experiment! I assume Fig. S1 should have been confused with Fig. S2 of the Supplemental Data S2. The authors need to make more clarity to guide the readers to explicative Figures better.

- Second paragraph of Results lines 164-166: the results presented in this part deserve a citation (potentially it is an article already cited before or later in the text).

- The closing sentence at the end of the second paragraph of Results (lines 167-169) can be misleading and need to be re-written. Maybe it might be acknowledged that in panel 2B MCL1 reduction is not that evident.

- Figure 1: Panel A; Given it represents cellular viability, it would have been more informative to express everything related to DMSO (set at 100%) and plotted as %. This may also help the reader to visualize IC50.

- From the Legends to Figure 2 it is not indicated that PTX was added alone and the combination treatment is not mentioned as well. Please adjust this.

- Figure 3B: MDA-MB-231 is not the best model to study cellular behavior in 3D. As it is possible to appreciate from the images, spheres are very dishomogenous, not well shaped, and spheres' roundness is limited. It is clear the authors did this additional experiment with this cell line to be consistent with the other results, but they need al least to mention this.

- From the legend of Figure 3B right panel, it is not clear what the luciferase activity refers to; and at what time point it was tested... day 6 or day 12?

---

## Round 0.2 · Minor Revisions

Please follow the indications given by the referee.

·

Basic reporting

No comment.

Experimental design

No comment.

Validity of the findings

No comment.

Additional comments

The manuscript has been revised according the suggestions of the reviewers, some novel additional experiments have been included, and it has gained more clarity and consistency. The Figure Legends have been improved and in the current version the reader can better appreciate the results from them.

However, there are some minor issues that need to be fixed prior final acceptance.

1. There are still some typos, particularly in the parts with new text. Please check this carefully.
2. Although in the rebuttal letter the authors mentioned they agreed with my comments for the seventh minor issue (about lines 167-169 of the original version) and have corrected the sentence, in the current version the sentence is still the same, it is still convoluted and the comment on MCL1 has not been included. The authors are requested to do it accordingly.
3. Although the authors agreed with my eight minor issue (Figure 1A presentation), and set the control condition (DMSO) to 100% as suggested (for viability), the axis name is still reporting Luciferase (RLU). Please modify this.
4. The new version of Figure 3 with the additional data on SUM159 needs to be adjusted. In my opinion, panel 3G and panel 3I can be reduced in size and the panel numbering changed as follows: current 3I can become 3H, and the two quantification plots respectively 3J (MB231 cells) and 3I (SUM159). In this way the numbering will be more linear.
5. In the Supplementary files, there are two pieces of data which seem unrelated to this story, or not requested for this submission. Raw data of a gene expression profile with the compounds used in the manuscript (excel table) and a heatmap summarizing this. What does these data refer to? Why are they not included in the original submission? Are there simply for a mistake and they will be part of a follow-up study? Please clarify at least the editor.

---

## Round 0.3 · accepted · Accept

The authors have adequately addressed all issues raised by the referees.